# Alteration of the Dopamine Receptors’ Expression in the Cerebellum of the Lysosomal Acid Phosphatase 2 Mutant (Naked–Ataxia (*NAX*)) Mouse

**DOI:** 10.3390/ijms21082914

**Published:** 2020-04-21

**Authors:** Mehdi Mehdizadeh, Niloufar Ashtari, Xiaodan Jiao, Maryam Rahimi Balaei, Asghar Marzban, Farshid Qiyami-Hour, Jiming Kong, Saeid Ghavami, Hassan Marzban

**Affiliations:** 1Cellular and Molecular Research Center, Department of Anatomy, Faculty of Medicine, Iran University of Medical Sciences, Tehran 1449614535, Iran; mehdizadeh.m@iums.ac.ir (M.M.); gheyamihor.f@tak.iums.ac.ir (F.Q.-H.); Jiming.Kong@umanitoba.ca (J.K.); Saeid.Ghavami@umanitoba.ca (S.G.); 2Department of Human Anatomy and Cell Science, The Children’s Hospital Research Institute of Manitoba (CHRIM), Max Rady College of Medicine, Rady Faculty of Health science, University of Manitoba, Winnipeg, MB R3E 0J9, Canada; ashtarin@myumanitoba.ca (N.A.); jiao.xiaodan@umanitoba.ca (X.J.); rahimibm@myumanitoba.ca (M.R.B.); 3Department of Pediatrics, School of Medicine, Zanjan University of Medical Sciences, Zanjan 4513956111, Iran; dmarzban@gmail.com; 4Research Institute in Oncology and Hematology, Cancer Care Manitoba, University of Manitoba, Winnipeg, MB R3E 0J9, Canada

**Keywords:** dopamine receptors, Purkinje cells, cerebellum, autophagy, *Acp2*

## Abstract

A spontaneous mutation in the lysosomal acid phosphatase (*Acp2*) enzyme (*nax*: naked–ataxia) in experimental mice results in delayed hair appearance and severe cytoarchitectural impairments of the cerebellum, such as a Purkinje cell (PC) migration defect. In our previous investigation, our team showed that Acp2 expression plans a significant role in cerebellar development. On the other hand, the dopaminergic system is also a player in central nervous system (CNS) development, including cerebellar structure and function. In the current investigation, we have explored how *Acp2* can be involved in the regulation of the dopaminergic pathway in the cerebellum via the regulation of dopamine receptor expression and patterning. We provided evidence about the distribution of different dopamine receptors in the developing cerebellum by comparing the expression of dopamine receptors on postnatal days (P) 5 and 17 between *nax* mice and wild–type (wt) littermates. To this aim, immunohistochemistry and Western blot analysis were conducted using five antibodies against dopamine receptors (DRD1, –2, –3, –4, and –5) accompanied by RNAseq data. Our results revealed that DRD1, –3, and –4 gene expressions significantly increased in *nax* cerebella but not in wt, while gene expressions of all 5 receptors were evident in PCs of both wt and *nax* cerebella. DRD3 was strongly expressed in the PCs’ somata and cerebellar nuclei neurons at P17 in *nax* mice, which was comparable to the expression levels in the cerebella of wt littermates. In addition, DRD3 was expressed in scattered cells in a granular layer reminiscent of Golgi cells and was observed in the wt cerebella but not in *nax* mice. DRD4 was expressed in a subset of PCs and appeared to align with the unique parasagittal stripes pattern. This study contributes to our understanding of alterations in the expression pattern of DRDs in the cerebellum of *nax* mice in comparison to their wt littermates, and it highlights the role of *Acp2* in regulating the dopaminergic system.

## 1. Introduction

The cerebellum has a crucial function in coordinating motor output, adapting involuntary reflexes, and regulating cognitive function [1,2,3]. Any alteration in the cerebellar structure and/or function, both during development and beyond, can contribute to a variety of movement and neuropsychiatric disorders and developmental syndromes [4,5,6]. Several of these disorders are associated with alterations in lysosomes and their acid hydrolases [7,8,9].

Lysosomes are membrane–bound organelles providing a perfect acidic pH (pH < 5) for degrading macromolecules [10,11,12]. Lysosomes contain over 50 different soluble acid hydrolases [13,14], such as lysosomal acid phosphatase 2 (ACP2), that play a pivotal role in the development of the cerebellum. ACP2 is commonly expressed in all mice organs, including the brain and cerebellum [9]. In the cerebellum, ACP2 is strongly expressed in cerebellar Purkinje cells (PCs) [9,15]. The significance of proper lysosomal function is disclosed by several human diseases, including neurodevelopmental and degenerative diseases, that occur as a result of mutations in lysosomal enzymes [7,16]. Mice that have experienced a targeted disruption of *Acp2* exhibit a mild phenotype with generalized lysosomal storage in the kidneys and central nervous system (CNS) [17]. A spontaneous autosomal recessive point mutation in the *Acp2* gene (*nax*: naked–ataxia) leads to several phenotypic abnormalities, weight loss, skin malformations, decreased size of the brain, and underdeveloped cerebellum, including cerebellar cortex disorganization with excessive PC migration and the disrupted compartmentation organization [16,18].

In the CNS, dopamine plays a crucial role in motor and cognitive functions [19,20], but whether the extent of the expression of dopamine receptors (DRDs) in the cerebellum regulates these functions is unclear. It has been shown that ACP2 is associated with progressive supranuclear palsy (PSP), which is an atypical parkinsonian condition [21]. Symptoms of PSP appear to originate from dysfunctions of multiple neurotransmitter systems, including dopaminergic systems [22,23,24,25].

Although the cerebellum has a unique role in movement and posture control, it was considered to be a nondopaminergic region for many years. Recently, it has been reported that dopaminergic projections to the cerebellar cortex and nuclei originate mostly from the ventral tegmental area (VTA), and most DRDs are present in PCs, which probably affects PC plasticity [26,27,28]. Implementation of anterograde and retrograde axonal tracing along with tyrosine hydroxylase immunofluorescence histochemistry revealed that the VTA in rat dispatches dopaminergic fibers that terminate in the granule and the Purkinje cell layer (gcl and Pcl, respectively) [26,27]. A comparable study between lurcher mice with 100% loss of cerebellar Purkinje cells and wild–type (wt) control mice demonstrated that, on the contrary to what happened in wt mice, cerebellar stimulation failed to evoke prefrontal cortex dopamine efflux in lurcher mice, indicating a dependency on cerebellar PC outputs [29]. According to the available data, dopamine–active transporter (DAT) immunoreactive axons present in specific lobules of the cerebellar vermis in primates primarily innervate the gcl and the adjacent Pcl [30]. Studies have illustrated that the cerebellum expresses all dopamine receptors (DRD1–5) [20,31,32]. In mice, DRD1 and –5 proteins have shown to be expressed in PCs and the molecular layer (mL), DRD2, in all cerebellar cortical layers (ml, Pcl, and gcl); DRD3 mostly in the Pcl and ml of lobules IX and X; and DRD4 in the ml of the cerebellum [30]. Furthermore, the presence of mRNA for the dopamine receptor subtypes has been demonstrated in the cerebellum [31,32,33]. One point worth mentioning is that *nax* mice exhibit mild to severe alterations in their gene expression patterns in the developing cerebellum [16], displaying the potential alteration of DRDs’ genes expression patterns in the cerebellum.

This study aims to compare the expression levels of DRDs in the cerebellum of *nax* mice and their wt control littermates in order to identify the effects of ACp2 mutation on the dopaminergic system and, specifically, its contribution to cerebellar development. Our findings provide novel insights on the differential expression of DRDs during the cerebellar development of mice.

## 2. Materials and Methods

### 2.1. Animal Maintenance

All animal procedures were performed in accordance with institutional regulations and the *Guide to the Care and Use of Experimental Animals* from the Canadian Council for Animal Care and has been approved by local authorities, “the Bannatyne Campus Animal Care Committee”, University of Manitoba (approved protocol #15066). A colony of *nax* mutant mice was established in the Genetic Model Center at the University of Manitoba by breeding mice (C57BL/6) heterozygous for the *nax* mutation (homozygote/heterozygote/wt ratio was ∼25%:50%:25%, respectively). Animals were grouped (*n* = 8 for each group) in standard polycarbonate cages with dust–reduced wood shavings as bedding. Animals were housed in groups of three per cage in a 12/12–h light/dark cycle (light on between 7:00 a.m. and 7:00 p.m.) with free access to chow and tap water. The animals were randomly allocated to different experimental groups.

### 2.2. Immunohistochemistry (IHC)

Peroxidase immunohistochemistry were carried out on cerebellar sections according to our lab protocol, as described previously [18,34]. Briefly, at two postnatal ages of P5 or P17, animals were transcardially perfused with 10–20 ml of 4% paraformaldehyde in 0.1–M phosphate buffer, pH 7.4. Brains were then dissected free and post–fixed overnight in the same fixative and cryoprotected with 10%, 20%, or 30% sucrose in the optimal cutting temperature (O.C.T.) compound. Free–floating sections were processed for light microscopic level studies, as described previously. All antibodies were diluted in 0.1–M phosphate–buffered saline (PBS, Gibco, Billings, MT, USA) with 10% normal goat serum and 0.3% Triton X–100. Sections were exposed overnight to overnight with primary affinity–purified antibody to one of the dopamine receptors (DRD1, DRD2, DRD3, DRD4, or DRD5 (1:1000)), followed by washing and secondary antibody exposure (goat anti–rabbit IgG or biotinylated goat anti–rabbit antibody (1:200; Jackson, West Grove, PA, United States). The reaction was developed by using either 0.05% diaminobenzidine or 0.01% hydrogen peroxide reaction.

### 2.3. Primary Antibodies Used for IHC and WB Analysis

Primary antibodies used to detect the five dopamine receptors. D1: rabbit polyclonal anti–D1 dopamine receptor (TA328798, anti–Drd1, diluted 1:1000; OriGene Biotech Co., Rockville, MD, USA); D2: rabbit polyclonal anti–D2 dopamine receptor (TA328800, anti–Drd2, diluted 1:1000; OriGene Biotech Co., Rockville, MD, USA), produced against recombinant rat dopamine receptor 2 (DR2); D3: rabbit polyclonal anti–D3 dopamine receptor (TA328800, anti–Drd3, diluted 1:1000; OriGene Biotech Co., Rockville, MD, USA), produced against recombinant rat dopamine receptor 3 (DR3); D4: rabbit polyclonal anti–D4 dopamine receptor (TA321202, anti–DRD4, diluted 1:1000; OriGene Biotech Co., Rockville, MD, USA), produced against recombinant rat dopamine receptor D4 (DRD4); and D5: rabbit polyclonal anti–D5 dopamine receptor (TA328802, anti–Drd5, diluted 1:1000; OriGene Biotech Co., Rockville, MD, USA), produced against recombinant rat dopamine receptor 5 (DR5).

### 2.4. Protein Extraction and Western Blot Analysis

The animals were sacrificed at P5 or P17, and cerebella were extracted. The following steps were performed to evaluate expression levels of the dopamine receptors (DRD1, DRD2, DRD3, DRD4, and DRD5). After sonication of the samples, protein concentration was measured by using a BCA protein assay kit (cat#5000121, Bio–Rad, Hercules, CA, USA). The protein sample was suspended in loading buffer [Tris–HCl 60mM, glycerol 25%, SDS 2%, mercaptoethanol (ME), (Thermo Fischer Scientific, Mississauga, ON, Canada)] 14.4mM, bromophenol blue 0.1%, and H_2_O) and separated by sodium dodecyl sulfate polyacrylamide gel electrophoresis. Based on the molecular weight of the protein, an 8%–15% polyacrylamide gel was used for electrophoresis. Twelve microliters of each sample was loaded, and 5–μl precision plus protein was used as the marker standard (Thermo Fischer Scientific, Mississauga, ON, Canada). The protein was then transferred to a polyvinylidene fluoride membrane. Then, blots were incubated with blocking solution, followed by overnight incubation with the primary antibody (see previous paragraph for details). After being washed, blots were incubated with horse radish peroxidase–conjugated antibody, and finally, immunoreactivity of polypeptides was detected using an enhanced chemiluminescence (ECL) solution. Quantification of the results was performed by densitometry (bands were scanned and analyzed by ImageJ software). All bands were normalized to β–actin expression.

### 2.5. RNA Isolation and Analyses

The cerebella of *nax* (P5, *n* = 2 and P7, *n* = 3) and wt littermates (P5, *n* = 3 and P7, *n* = 3) were dissected, and total RNA was extracted using the RNeasy Plus Mini Kit (Catalog No. 74134, QIAGEN, Hilden, Germany). RNA product concentration was measured by Nano–Drop ND–1000 UV–Vis Spectrophotometer (Thermo Fisher Scientific, Waltham, MA, USA). The samples were kept at −80 °C and sent to the McGill University and Genome Quebec Innovation Centre (MUGQIC). Raw RNAseq data were analyzed with FASTQ, and files were filtered and trimmed using the fastp package [35]. Then, the cleaned data were aligned with the mouse reference genome (GRCm38.97) using the Subjunc Aligner [36]. Next, low–quality alignments, duplicates, and nonaligned sequences were removed from the aligned files using Samtools [25]. Finally, the number of reads associated with each gene on the reference genome was extracted and counted using FeatureCounts software [37], and the Reads Per Kilobase of transcript per Million mapped reads (RPKM) was calculated for each gene.

### 2.6. Statistical Analysis

GraphPad Prism 7.05 was used to analyze the difference by two–way analysis of variance (ANOVA) for more than two groups and by a Student’s *t*–test for comparison of the two groups. Differences were considered to be statistically significant when *p* < 0.05 and are indicated in the figures as *(*p* < 0. 05) and **(*p* < 0.01).

### 2.7. Imaging

Images were captured by using a Zeiss Axio Imager M2 microscope (Zeiss, Toronto, ON, Canada). Images were cropped, corrected for brightness and contrast, and assembled into montages using Adobe Photoshop CS5 Version 12.

## 3. Results

In order to understand whether the DRDs are altered, RNA sequencing was performed in *nax* (*n* = 5) and wt (*n* = 6) cerebellum at P5 and P7 (P5/P7). RNAseq at P5/P7 revealed Drd1, Drd3, and Drd4 were significantly increased in the *nax* cerebellum compared to wt cerebellum. Although Drd2 and Drd5 expression increased in the *nax* cerebellum, it was not statistically significant from the wt littermates (Figure 1).

### 3.1. Dopamine Receptor D1 (DRD1) Expression in wt and nax Cerebella at P5 and P17

Immunoperoxidase staining of cerebellar sections for DRD1 in wt (Figure 2A) and *nax* mice (Figure 2B) was compared at P5. Immunoreactivity was observed primarily in the cerebellar cortex and white matter. DRD1 in the *nax* cerebellum showed similar expression as in the *wt* littermates, and immunoreactivity was seen in the Purkinje cell layer (Pcl) and white matter (wm) but not in the external germinal zone (egz) and granular layer (gl). By P17, DRD1 immunoreactivity was broadly present in all three layers of the cerebellar cortex, particularly in the gl, PC somata in the Pcl, and in dendrites residing in the molecular layer (mL), but not in the wm of wt controls (Figure 2C). DRD1 was primarily expressed in PC somata in the Purkinje cell/molecular layer (Pcl/mL; referred to as such, because PCs have invaded into the mL) but not as much in the gl (as compared to wt) in the *nax* cerebellum at P17 (Figure 2D). DRD1 immunoreactivity was weakly detected in cerebellar nuclei neurons in the *nax* cerebellum, which was similar to the wt littermates at P17 (data not shown). Western blot analysis revealed no significant differences in total cerebellar DRD1 protein expression between both groups at P5 and P17 (Figure 2E). Interestingly, RNAseq at P5/P7 revealed a significant upregulation in *Drd1* RNA transcript in the *nax* as compared to the wt cerebellum (Figure 1A).

### 3.2. Dopamine Receptor D2 (DRD2) Expression in wt and nax Cerebella at P5 and P17

Immunohistochemistry using a DRD2 antibody showed immunoperoxidase reactivity in the cerebellar cortex on both P5 and P17. By P5, DRD2 is weakly expressed in Pcl and egz of the wt littermate cerebellum (Figure 3A)*,* whereas a relatively strong expression was observed in PC somata of the *nax* cerebellum (Figure 3B). By P17, DRD2 expression in the cerebellar cortex was localized mostly in PC somata and the ml but not in the gcl, which was comparable between wt (Figure 3C) and *nax* (Figure 3D) cerebella. Western blot analysis was performed at P5 and P17 using lysates of total wt and *nax* cerebella, and it showed a trend towards a slight increase in DRD2 expression, although these differences did not reach significance (Figure 3E). Similarly, RNAseq at P5/P7 did not reveal a significant increase in *Drd2* gene expression in the *nax* cerebellum; thus, there were no significant differences in DRD2 expression between *nax* and wt cerebella (Figure 1B).

### 3.3. Dopamine Receptor D3 (DRD3) Expression in wt and nax Cerebella at P5 and P17

Immunoperoxidase staining for DRD3 revealed weak expression in the cerebellar cortex at P5 both in wt (Figure 3A) and *nax* (Figure 4B) cerebella. Whereas, at P17, DRD3 immunoreactivity was detectable in the whole cortex of the wt cerebellum and was strongly and uniformly expressed in PC somata (Figure 4C,D). In addition, DRD3 immunoreactivity was detected in scattered cell bodies, similar in appearance to Golgi cells, in the gl (indicated arrows in Figure 4D(d)). Cerebellar nuclei neurons (CNs) were strongly immunopositive to DRD3 (Figure 4C,E). In the *nax* cerebellum at P17, strong DRD3 expression was observed in the cerebellar cortex and CNs (Figure 4F). In addition, DRD3 was present in multilayered PC somata in the Pcl/ml (due to excessive PC migration to mL) (Figure 4F,G), whereas there was no or only very little scattered immunoreactivity observed in the cells similar to Golgi cells in the gl (Figure 4G,H). Western blot analysis showed comparable DRD3 expressions in wt and *nax* cerebella at P5 and P17 (Figure 4I). Interestingly, however, RNAseq at P5/P7 revealed that *Drd3* gene expression was significantly higher in *nax* than in wt cerebella (Figure 1C).

### 3.4. Dopamine Receptor D4 (DRD4) Expression in wt and nax Cerebella on P5 and P17

At P5, weak DRD4 expression was detectable in the cerebellar cortex, mostly in the egz, of both wt (Figure 5A) and *nax* (Figure 5B) cerebella. Whereas, at P17, DRD4 showed strong immunoreactivity in the entire cerebellar cortex of wt mice (Figure 5C–G). In sagittal sections of the wt cerebellum, PC somata immunoreactivity was strong in the nodular zone (NZ) but was lacking in the anterior zone (AZ), while PC dendrites were immunopositive (Figure 5C). This pattern of gene expression was aligned with the cerebellar zone and stripe cytoarchitecture. The cerebellum can be divided into four transverse zones based on gene expression, afferents, and efferents: the anterior zone (AZ: lobules I–V), the central zone (CZ: lobules VI–VII), the posterior zone (PZ: lobules VIII + dorsal lobule IX), and the nodular zone (NZ: ventral lobule IX + lobule X). Additionally, each zone is subdivided into parasagittal stripes (e.g., [15,16,18,38,39]). In the frontal section of the wt cerebellum, DRD4 expression was strongly present in a subset of PCs (Figure 5D, black arrow heads), while no or only weak expression was detectable in some of the other PCs (Figure 5D, arrow and white arrow head, respectively) at P17. In addition, DRD4 expression in the wt cerebellum displayed a stripes pattern in lobules VIII and IX (Figure 5E,F), while expression was uniform in PCs of lobule X (Figure 5G). In the *nax* cerebellum at P17, DRD4 showed immunoreactivity in a subset of PCs (indicated by black asterisks (Figure 4H)), which was alternated with a weak or lack of DRD4 expression similar to the stripes pattern in lobule III (Figure 5H, white asterisk). The same pattern was present in lobule IX (Figure 5I), whereas DRD4 was uniformly expressed in PCs of lobule X (Figure 5J). Western blot analysis was performed at P5 and P17 using lysates of total wt and *nax* cerebella, and it showed a slight increase in DRD4 expression, although these differences did not reach significance (Figure 5K). As for *DRD1* and *–3*, RNAseq analysis at P5/P7 revealed that *Drd4* gene expression was significantly higher in *nax* than in wt cerebella (Figure 1D).

### 3.5. Dopamine receptor D5 (DRD5) expression in wt and nax cerebella on P5 and P17

Immunoperoxidase staining for DRD5 showed weak immunoreactivity in the Pcl of both wt (Figure 6A) and *nax* (Figure 6B) cerebella at P5. A similar weak expression pattern in the Pcl, with expression primarily localized in PCs, was observed at P17 for both groups (Figure 6C,D). Western blot analysis was performed at P5 and P17 using lysates of total wt and *nax* cerebella and revealed a slight increase in DRD5 expression, although these differences did not reach significance (Figure 6E). Similarly, RNAseq at P5/P7 showed a relative increase in *Drd5* gene expression in the *nax* cerebellum, which was not significant (Figure 1E).

## 4. Discussion

In this study, we used postnatal *nax* mice to determine whether anomalies of the cerebellar cortex and PC degeneration and excessive migration are associated with alterations in the expression patterns of various DRDs. Overall, the expression of DRDs in early postnatal development was dynamic, from very weak and mostly localized in the Pcl (within the PC somata) and external germinal zone (egz) to more strongly towards the whole cerebellar cortex, in which PCs express DRDs uniformly—for instance, as for DRD3 or, in the characteristic parasagittal stripes pattern, as for DRD4. Despite several apparent abnormalities and loss and excessive migration of PCs in the *nax* cerebellum, there seemed to be no obvious (significant) differences in protein expression of the DRDs between *nax* and wt littermate cerebella, especially when studying total cerebellar lysates. However, significant differences were observed in gene expression levels of *Drd1*, *Drd3*, and *Drd4* at P5/P7. The pattern of the *Drd3* and *Drd4* expression appeared to be associated with the cerebellar cytoarchitecture changes during postnatal development.

There is a significant body of anatomical evidence to support the existence of a direct connection between the cerebellum and the dopaminergic mesencephalon [26,30]. Dopaminergic fiber projection from the VTA to the cerebellar cortex, which is mainly terminated in the gl and Pcl, has been demonstrated [27]. In addition, quantitative analyses revealed that DAT, which actively translocates released dopamine from the synaptic cleft into the presynaptic neurons [40], is present in the cerebellum and, in particular, in higher levels in the ml and cerebellar nuclei [41], which in itself is a sign that dopamine and dopamine receptors exist in these sites. Based on their structural characteristics, dopamine receptors are divided into two groups: D1–like (DRD1 and –5) and D2–like (DRD2, -3, and -4) [42]. In the rat cerebellum, DRD1 and DRD5 were localized in PCs and the ml, DRD2 in all layers, DRD3 mainly in the PCs and the ml of lobules IX and X, and DRD4 in the ml [30]. Another study claimed that, in rat cerebellum, DRD1 was found only in PCs and the ml and DRD2 exclusively in the ml of lobules IX and X [43]. These findings indicate that the exact expression profile of the different DRDs in the cerebellum is still up for debate and requires further elucidation.

DRD1 is involved in mediating neuronal plasticity and has a prominent role in cognitive functions, including learning and memory and spatial navigation [44,45,46]. Our results demonstrated that DRD1 was expressed in both the wt and *nax* cerebellar cortex. It is of particular interest to mention that, at P5, no immunoreactivity at all was observed in the ml but that the PC somata and dendrites in the ml were immunoreactive at P17. This suggests that DRD1 is synthesized in the gl and Pcl and extended toward the ml on PC dendrites during cerebellar development. Autoradiographic combined with in situ hybridization techniques demonstrated that dopamine receptor binding, especially for the *Drd1* subtype, occurs in both the ml and gl [27], while dopamine receptor mRNAs for both DRD1 and DRD2 subtypes are only present in the gl [27]. Thus, this implies that cerebellar dopamine receptors are synthesized in the gl, perhaps in granule cell bodies, and are either transported through granule cell axons to the ml to form presynaptic receptors or remain in granule cell bodies to constitute postsynaptic receptors [27].

Rani and Kanungo claimed that the expression of DRD2 occurs at birth in both cerebral and cerebellar cortices of mice. However, in the cerebellar cortex, a high expression of D2DR has been reported at birth (at P0) [47]. Researchers also suggested that the *Drd2* gene has a role in neuronal maturation and dopaminergic synapse formation up to P15 of postnatal development of the murine brain, during which the brain is still developing [47]. Paradoxically, our study indicates that, despite the severe impairment in cerebellar cortex in *nax* mice, DRD2 expression is similar as in wt littermates and that the neuronal maturation and synapse formation defects have no direct effect on DRD2 expression profiles. This does not exclude a functional role for DRD2 in neuronal maturation—or any other developmental processes, for that matter; it simply indicates that expression levels are unaltered under these specific conditions, which does not necessarily reflect receptor function or ligand interaction dynamics.

A controversial functional aspect of DRD3 is its role as an autoreceptor, regulating the dopaminergic activity of neurons [48]. In our study, although PC somata and cerebellar nuclei neuronal bodies were strongly immunoreactive at P17, expression of DRD3 was very weak in both wt and *nax* at P5. It has been shown that, in the Pcl, PCs expressed *Drd3* mRNA, whereas dopamine binding sites were found in the ml, where the dendritic trees of PCs are present [49,50]. Our results showed that, both in the wt and *nax* cerebellum, DRD3 is strongly expressed in PCs and cerebellar nuclei neuronal bodies but not in PC dendrites at the sites of the synapses with parallel fibers. This is particularly evident in the *nax* cerebellum, in which, due to the dramatically decreased number of granule cells/parallel fibers (the manuscript of this study is under review), PC somata are highly immunopositive for DRD3, suggesting they autonomously express DRD3. In fact, RNAseq data show that the *Drd3* gene expression is significantly upregulated in the *nax* cerebellum. Furthermore, Golgi cells in the gl highly expresses DRD3 but is absent or very weak in *nax* cerebellum. Therefore, this suggests that DRD3 may contribute to a cerebellar cortex organization during postnatal corticogenesis as well.

Studies have reported the expression of DRD4 in the ml, in PCs, in stellate neurons of the ml, in a few Golgi neurons of the gl, and in structures resembling climbing fibers [51,52]. In our current study, DRD4 protein expression was slightly increased in *nax* mice cerebella. Despite the relatively weak DRD4 expression at P5, which was localized in the egz both in wt and *nax* cerebella, strong immunoreactivity was evident in a subset of PC somata and dendrites at P17. The DRD4 expression profile in PC somata and/or dendrites was different between the cerebella from wt and *nax* mice. For instance, only the PC dendrites, but not somata, in the AZ are immunoreactive, whereas both PC somata and dendrites were immunoreactive in the NZ. Furthermore, there were striped expression patterns in lobules VIII and IX (i.e., PZ), whereas a uniform expression was evident in PCs of lobule X (i.e., NZ). These patterns of zone and stripes are clearer at P17 in the *nax* cerebellum. DRD4 immunoreactivity of PCs displays the stripes pattern of expression in lobules III and IX but a uniform expression in lobule X in *nax* cerebella. This indicates that DRD4 is expressed in a stripe pattern in the wt cerebellum, but the clear appearance of a stripes pattern in *nax* cerebellum, particularly in the AZ, is probably because of the decreased number of stripes in *nax* mice [16]. It has been reported previously that only a faint DRD4 immunoreactivity is present in the ml, associated with structures resembling climbing fibers, PCs, and stellate neurons of ml [53].

In our study, DRD5 protein and gene expression patterns during cerebellar development were comparable between the wt and *nax* group. Immunoperoxidase staining on P5 showed a weak immunoreactivity of DRD5 in the Pcl in wt and *nax* cerebella. By P17, DRD5 immunoperoxidase staining showed weak immunoreactivity in PCs in the wt and *nax* cerebellum. A study by Barili and colleagues showed a faint DRD5 immunoreactivity in the cerebellar white matter of rats, suggesting these receptors may be presynaptic and transported through axons to the cortical layers [53].

In a final note, it is important to discuss the possibility of the involvement of the cell recycling process, macroautophagy (hereafter, autophagy), and mechanism of the cell in response to cellular stress, unfolded protein response, and its possible involvements in our observations. Acp2 is a lysosomal enzyme which might be involved in the regulation of lysosomal activity [54] and a possible indirect effect in the regulation of autophagy. It is a strong possibility that mutated Acp2 can potentially affect the autophagy flux and the trafficking of the protein degradation by lysosomes (e.g., Appendix A). Our results showed that ACP2 point mutation can not only affect autophagosome recycling (LC3β lipidation) but also can affect expression of the essential proteins which are involved in different stages of autophagy Beclin–1, Atg7). As a result, we potentially have the defect of lysosomal protein degradation in *nax* animals due to a malfunction of lysosomal activity. Therefore, it is possible that our observations regarding the mismatch of RNAseq results and protein expressions of DRDs were due to a defect of autophagy and lysosomal protein degradation in the *nax* model. On the other hand, our recent investigations have showed that unfolded protein response UPR is a regulator of autophagy in many models [55,56,57]. It is very probable that mutated/misfolded Acp2 induced UPR and indirectly affected the autophagy and lysosomal activity and downregulated the lysosomal protein degradation and, subsequently, downregulated the DRD degradation. At present, my team is focused on the regulation of UPR and autophagy in a *nax* model and its possible effects in intracellular trafficking and affecting early development.

## 5. Conclusions

In conclusion, our comparative study on dopamine receptor subtype expression in the developing cerebella of *nax* and wt mice showed no significant differences in DRD1–5 protein expression levels but did reveal significant upregulations in *Drd1*, *Drd3*, and *Drd4* gene expression in *nax* cerebella. It seems that *Drd2* and *Drd5* gene expression have no dependency on ACP2. With regard to differential DRD expressions, the most interesting observation included the lack of or very weak DRD3 expression in Golgi cells and the more pronounced DRD4 stripe expression pattern in the *nax* cerebellum. Alterations in cerebellar DRD expression levels, most noticeably in DRD3 and –4, may have resulted from an *Acp2* mutation and may therefore play a role in cerebellar development and the control of movement.

## Figures and Tables

**Figure 1 ijms-21-02914-f001:**
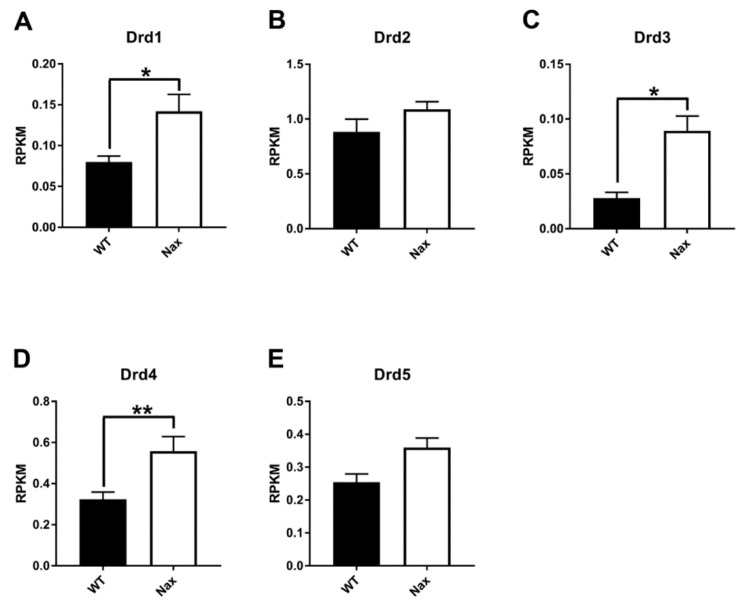
RNA sequencing data analysis in *nax* (*n* = 5) and wild–type (wt) (*n* = 6) mice. Data analysis shows dopamine receptor D1 (*Drd1,*
**A**), *Drd3* (**C**), and *Drd4* (**D**) are significantly increased in the *nax* cerebellum. Although an increase in *Drd2* (**B**) and *Drd5* (**E**) is apparent, statistical significance was not reached. The data in the bar graph are presented as the mean ± SEM, and statistical analysis was performed using an unpaired *t*–test (* *p* < 0.05 and ** *p* < 0.01). RPKM: Reads Per Kilobase of transcript per Million mapped reads.

**Figure 2 ijms-21-02914-f002:**
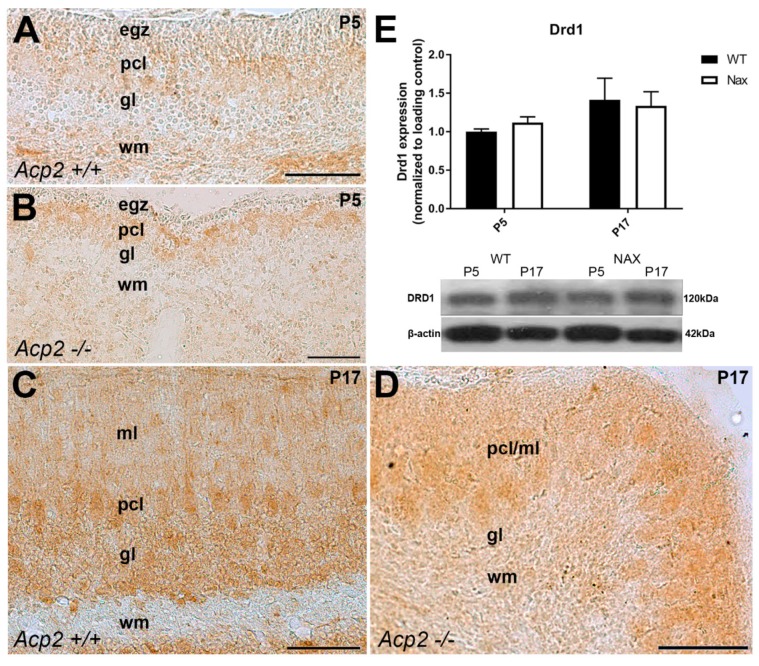
Frontal sections of wt and *nax* mouse cerebella: DRD1 expression at P5 and P17. (**A**) Immunoperoxidase staining for DRD1 in the wt cerebellum on P5 shows the immunoreactivity in the Purkinje cell layer (Pcl) and white matter (wm) but not in the external germinal zone (egz) and granular layer (gl). (**B**) Immunoperoxidase staining for DRD1 in the *nax* cerebellum shows similar immunoreactivity as in the wt littermates at P5. (**C**) DRD1 immunostaining of the frontal sections of the wt cerebellum at P17 shows immunoreactivity in the Purkinje cell (PC) somata, the Pcl, and dendrites in the molecular layer (ml) and in the gl but not in the wm. (**D**) DRD1 immunostaining of the frontal sections of the *nax* cerebellum on P17 shows immunoreactivity in the PCs in the ml/Pcl. (**E**) Western blot analysis of whole cerebellar lysates revealed no significant differences in DRD1 expression between wt and *nax* cerebella at P5 and P17 (wt: *n* = 3 and *nax*: *n* = 3). The data in the bar graphs are presented as the means of three independent experiments ± SEM, and statistical analysis was performed using a two–way ANOVA. P; postnatal. Scale bars: 100 μm in A, B, C, and D.

**Figure 3 ijms-21-02914-f003:**
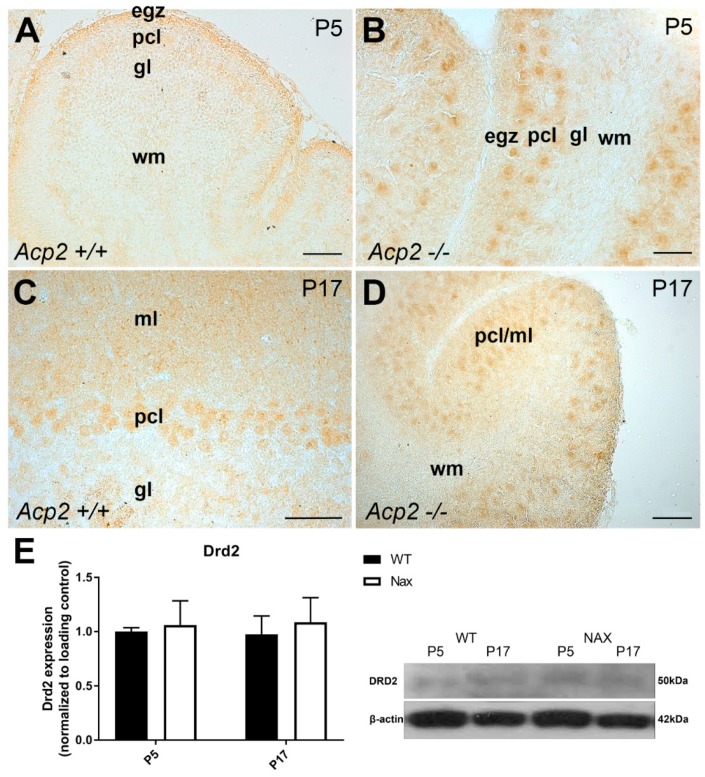
Frontal sections of wt and *nax* mouse cerebella: DRD2 expression at P5 and P17. Immunoperoxidase staining for DRD2 in wt (**A**) and *nax* (**B**) cerebella at P5 shows weak immunoreactivity in the Pcl of wt but relatively strong immunoreactivity in PC somata of the *nax* cerebellum. (**C**,**D**) Immunoperoxidase staining for DRD2 in wt (**C**) and *nax* (**D**) cerebella at P17 shows weak immunoreactivity in PCs, with no differences between the two groups. (**E**) Western blot analysis of whole cerebellar lysates revealed no significant differences in DRD2 protein expression between wt and *nax* cerebella at P5 and P17 (wt: *n* = 3 and *nax*: *n* = 3). The data in the bar graphs are presented as the means of three independent experiments ± SEM; statistical analysis was performed using two–way ANOVA. P; postnatal. Scale bars: 100 μm in A, C, and D and 50 μm in B.

**Figure 4 ijms-21-02914-f004:**
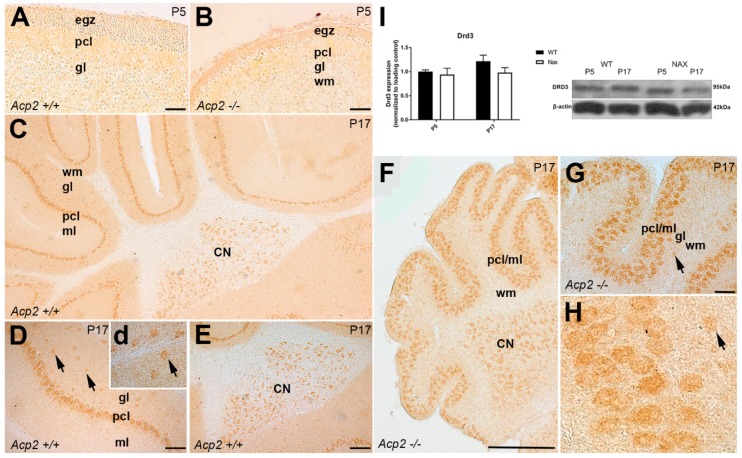
Frontal sections of wt and nax mouse cerebella: DRD3 expression at P5 and P17. (**A**,**B**) Immunoperoxidase staining for DRD3 in wt (**A**) and *nax* (**B**) cerebella at P5 shows weak immunoreactivity in the Pcl and external germinal zone (egz). (**C**–**E**) Immunoperoxidase staining for DRD3 in the wt cerebellum at P17 demonstrates immunoreactivity in the whole cortex of the wt cerebellum, mostly in PC somata (**C**,**D**) within scattered cell bodies in the gl (arrows indicated in **D**(**d**)) and in cerebellar nuclei neurons (CNs) (**C**,**E**). (**F**–**H**) DRD3 immunostaining of a frontal section of the *nax* cerebellum at P17 shows strong immunoreactivity in PC somata in the Pcl/ml (**F**–**H**), CNs (**F**), and a few cells in the gl (arrows in **G**,**H**). (**E**) Western blot analysis of whole cerebellar lysates revealed no significant differences in DRD3 protein expression between wt and *nax* cerebella at P5 and P17 (wt: *n* = 3 and *nax*: *n* = 3). The data in the bar graph are presented as the mean of three independent experiments ± SEM, and statistical analysis was performed using two–way ANOVA. P; postnatal. Scale bars: 50 μm in A and B, 500 μm in C and F, 100 μm in D, 200 μm in E, 100 μm in G.

**Figure 5 ijms-21-02914-f005:**
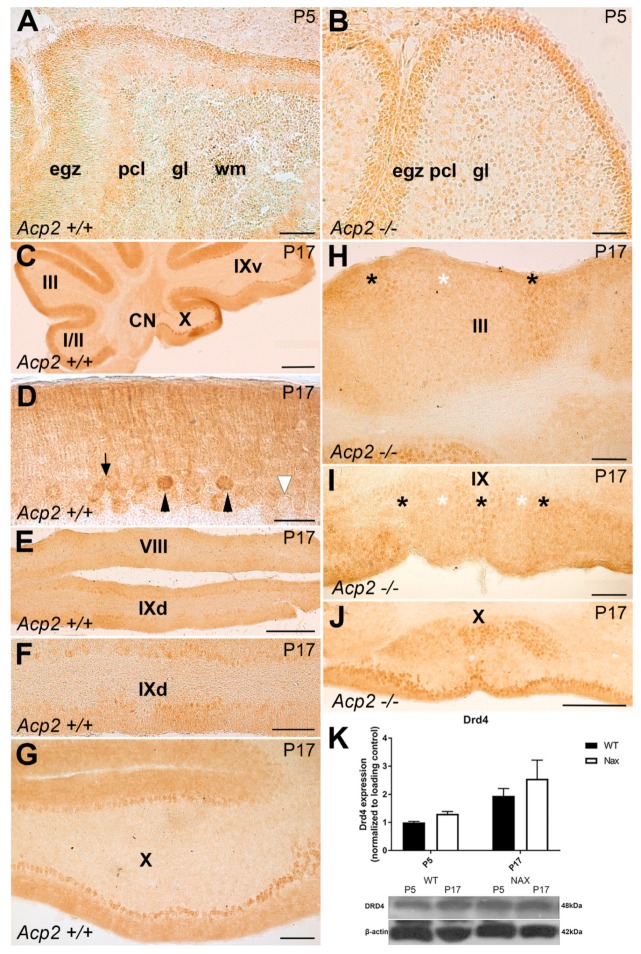
Frontal and sagittal sections of wt and *nax* mouse cerebella: DRD4 expression at P5 and P17. (**A**,**B**) Immunoperoxidase staining for DRD4 in wt (**A**) and *nax* (**B**) cerebella at P5 shows immunoreactivity in the external germinal zone (egz) and no or only weak immunoreactivity in the Pcl. (**C**–**G**) Sagittal section of P17 wt cerebellum immunostained for DRD4 reveals strong immunoreactivity within the cerebellar cortex. In the anterior zone, the expression of DRD4 is present in PC dendrites in the molecular layer (ml), while PC somata and dendrites exhibit immunoreactivity in the nodular zone (**C**). Frontal section of P17 wt cerebellum immunostained with DRD4 shows strong immunoreactivity in a subset of PCs (indicated by black arrowheads), while another subset of PCs displays no (white arrowhead) or only weak expression (arrow) in the cerebellar cortex (**D**). (**E**–**G**) Immunoperoxidase staining for DRD4 in the wt cerebellum at P17 shows a striped expression pattern in lobule IXd (**E**,**F**) and more uniform expression in PCs in lobule X (**G**). (**H**–**J**) DRD4 immunostaining of a frontal section of the *nax* cerebellum at P17 shows immunoreactivity in a subset of PCs (asterisks) that follows a similar stripes pattern in lobule III (**H**) and IX (**I**) but is uniform in lobule X (**J**). (**K**) Western blot analysis of DRD4 expression in wt and *nax* cerebellum at P5 and P17 shows an apparent increase in expression in *nax* cerebella at both time–points (wt: *n* = 3 and *nax*: *n* = 3); however, these differences did not reach statistical significance. The data in the bar graphs are presented as the means of three independent experiments ± SEM, and statistical analysis was performed using two–way ANOVA. P; postnatal. Scale bars: 100 μm in A; 50 μm in B; 500 μm in C; 20 μm in D; 200 μm in E, F, G, H, and I; and 500 μm in J.

**Figure 6 ijms-21-02914-f006:**
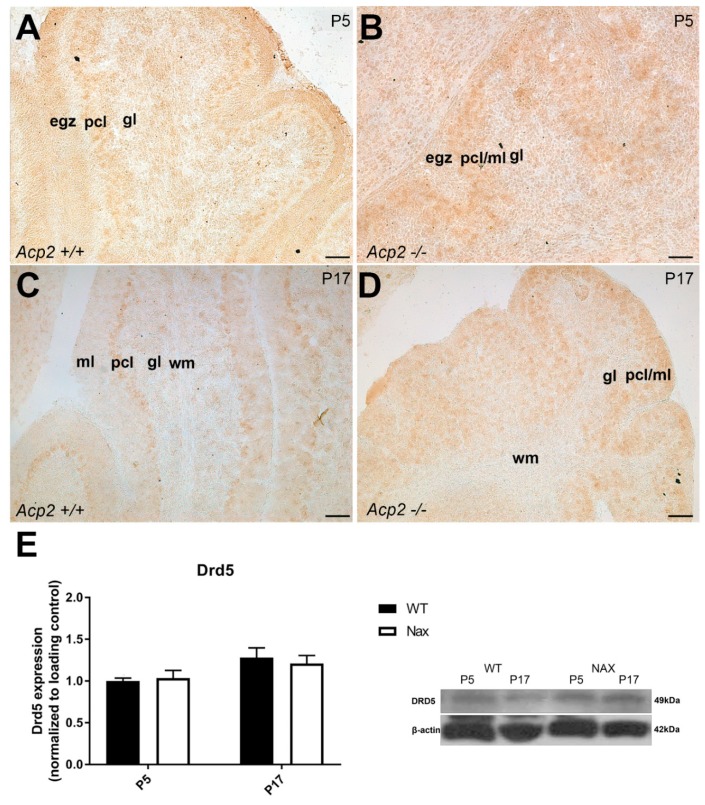
Sagittal sections of wt and *nax* mouse cerebella: DRD5 expression at P5 and P17. (**A**,**B**) Immunoperoxidase staining for DRD5 in wt (**A**) and *nax* (**B**) cerebella at P5 shows weak immunoreactivity in the Pcl. (**C**,**D**) Immunoperoxidase staining for DRD5 at P17 shows weak immunoreactivity in PCs in the wt (**C**) and *nax* (**D**) cerebellar cortex. (**E**) Western blot analysis of whole cerebellar lysates revealed no significant differences in DRD5 protein expression between wt and *nax* cerebella at P5 and P17 (wt: n = 3 and *nax*: n = 3). The data in the bar graphs are presented as the means of three independent experiments ± SEM, and statistical analysis was performed using two–way ANOVA. P; postnatal. Scale bars: 50 μm in A and B and 100 μm in C and D.

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
