# Peer review of "Alteration of the Dopamine Receptors’ Expression in the Cerebellum of the Lysosomal Acid Phosphatase 2 Mutant (Naked–Ataxia (NAX)) Mouse"

_ijms, 2020, doi:10.3390/ijms21082914_

Round 1

Reviewer 1 Report

Mehdizadeh and colleagues provided evidence about the distribution of different dopamine receptors (DRDs) in the developing cerebellum of lysosomal acid phosphatase 2 mutant (nax) mice. They compared the expression of DRDs on postnatal 5 and 17 between wt and nax cerebella. In particular, the Authors found that gene expression of all dopamine receptors (DRD1-DRD5) was evident in cerebellar cortex with some cell-type differences between nax and wt mice. Interestingly, DRD3 was expressed in Golgi cells in the wt but not in nax cerebella and DRD4 was epressed in a subset of Purknjie cells and appeared to align with the parasagittal stripes pattern. This study is nicely presented but I have some suggestions.

The introduction should be reorganized by including additional references (for example after the sentences “Several of these disorders are associated with alterations in lysosomes and its acid hydrolases” and “Dopaminergic projections to the cerebellar cortex ….. probably affects PC plasticity").

It is advisable to describe the previous results on the nax cerebellum in more detail (Bailey et al., 2014).

The reference 22 (He et al., 2014) is not appropriate. Please add some more accurate.

In my opinion the introduction will benefit from a different flow of the points discussed. Below I will provide a couple of suggestions.

After the description of the effects of the Acp2 gene mutation on the cerebellar cortex disorganization (Bailey et al., 2014), you could introduce the concept of the link between ACP2, Parkinsonian condition and dopaminergic system dysfunctions.
You could then continue with an overview about dopaminergic innervation and localization of DRD1-DRD5 receptors in the cerebellum.

Please standardize the calibration bars for all figures, where possible. Moreover, increase the font size. They are not very visible and therefore difficult to read.

In the figure 1, indicate the number of experiments (wt: n=?; nax: n=?).

The authors compared the cerebellar expression of dopamine receptors on P5/P17 nax and wt cerebella, but RNA sequencing was only performed at P5. It would be advisable to also add the RNA sequencing to P17 to verify if, during the development, the DRDs expression changes.

The staining for DRD1 in the nax cerebellum shows similar immunoreactivity as in the wt littermates at P5. This result is not clearly expressed in the paragraph 3.1. It would have to be rephrased.

The figure 6 shows the legend of results described in the paragraph 3.5. Moreover, the figure describing the results of the paragraph 3.5 is missing.

I would rename figure 6 to figure 3.

Page 7 line 205: “At P5; DRD2 is relatively strongly expressed in Purkinje cell layer in the nax cerebellum”. This sentence does not match the one written in the legend of figure 3.

Page 7 line 210-211 “Similarly, RNAseq at P5/P7 revealed an increase in DRD2 gene expression in the nax cerebellum” is not completely true. Please correct with “ did not reveal a significant increase”.

It appears from figure 4 that there are no differences in DRD3 expression between nax and wt mice in deep cerebellar nuclei. (CN). Since 2 papers (Cutando et al., 2019; Locke et al., 2018) have recently identified DRD1-DRD2-expressing neurons in CN, it could be useful to check whether the CN, in addition to expressing DRD3, also express the DRD1 and DRD2 and thus verify whether differences exist between nax and wt mice.

In the nax cerebellum, DRD4 is expressed in a subset of PCs which is alternated with weak or lack of DRD4 expression similar to parasagittal stripes pattern (Figure 5). It would be very interesting to perform double immunostaining with zebrinII in order to reveal a possible array of alternating ZII/DRD4 parasagittal stripes in nax and wt cerebella.

Author Response

Dr. Hassan Marzban

Department of Human Anatomy and Cell Science

College of Medicine, Faculty of Health Sciences

University of Manitoba

745 Bannatyne Avenue

Winnipeg, Manitoba, R3E 0J9

Canada

Editor-in-Chief

                                                                                                                    April 12, 2020

Dear Editor,

             We have revised the MS entitled:  “ALTERATION OF THE DOPAMINE RECEPTORS EXPRESSION IN THE CEREBELLUM OF THE LYSOSOMAL ACID PHOSPHATASE 2 MUTANT (NAKED-ATAXIA [NAX]) MOUSE” authored by Mehdi Mehdizadeh, Niloufar Ashtari, Xiaodan Jiao, Maryam Rahimi Balaei,  Asghar Marzban,  Farshid Qiyami Hour, Jiming Kong, Saeid Ghavami, and Hassan Marzban according to the reviewers' careful suggestions and comments. The details of the changes are as follows:

Reviewer #1

The introduction should be reorganized by including additional references (for example after the sentences “Several of these disorders are associated with alterations in lysosomes and its acid hydrolases” and “Dopaminergic projections to the cerebellar cortex ….. probably affects PC plasticity").

Answer:  Thank you for reviewer valuable comment, the introduction has been reorganized and more relevant references have been cited as following:   Jiao, 2017, Cataldo et al., 1994, Mannan et al., 2004 on page 1 and Ikai et al., 1992, Ikai et al., 1994 on page 2  highlighted in yellow.

It is advisable to describe the previous results on the nax cerebellum in more detail (Bailey et al., 2014).

Answer:  The main features from Bailey et al., 2014 has been described in the 2nd paragraph of the introduction. (Highlighted in yellow).

The reference 22 (He et al., 2014) is not appropriate. Please add some more accurate.

Answer:  According to reviewer valuable comment, the reference has been replaced with relevant reference and information (Mittleman G and et al., 2008) on page 2 and highlighted in yellow.

In my opinion the introduction will benefit from a different flow of the points discussed. Below I will provide a couple of suggestions.

After the description of the effects of the Acp2 gene mutation on the cerebellar cortex disorganization (Bailey et al., 2014), you could introduce the concept of the link between ACP2, Parkinsonian condition and dopaminergic system dysfunctions.

You could then continue with an overview about dopaminergic innervation and localization of DRD1-DRD5 receptors in the cerebellum.

Answer: The introduction has been reorganized and focused accordingly. The flow of introduction has been changed as following: (highlighted in yellow)

  • Cerebellum and being a relation between alteration in cerebellar structure and disorders
  • Effects of the Acp2 gene mutation on the cerebellar cortex disorganization
  • The link between ACP2, Parkinsonism and dopaminergic system dysfunctions
  • Dopaminergic innervation and localization of DRD1-5 receptors in the cerebellum
  • The aim of study

Please standardize the calibration bars for all figures, where possible. Moreover, increase the font size. They are not very visible and therefore difficult to read.

Answer:  Thank you, the calibration bars and the font size have been fixed.

In the figure 1, indicate the number of experiments (wt: n=?; nax: n=?).

Answer:  The number of experiments has been mentioned in the figure legend and highlighted in yellow.

The authors compared the cerebellar expression of dopamine receptors on P5/P17 nax and wt cerebella, but RNA sequencing was only performed at P5. It would be advisable to also add the RNA sequencing to P17 to verify if, during the development, the DRDs expression changes.

Answer:  We agree. This experiment, and related ones, is on our to-do list, but the data will not be available for this MS.

The staining for DRD1 in the nax cerebellum shows similar immunoreactivity as in the wt littermates at P5. This result is not clearly expressed in the paragraph 3.1. It would have to be rephrased.

Answer:  It  has been rephrased and clearly expressed in the paragraph 3.1. (highlighted in yellow)

The figure 6 shows the legend of results described in the paragraph 3.5. Moreover, the figure describing the results of the paragraph 3.5 is missing.

I would rename figure 6 to figure 3.

Answer:  According to reviewer comments, the relevant figure has been added and the necessary changes applied. To this end, the figures associated to Western blot bands were added next to its related analysis chart along with immunostained figures (applied to all figures). Figure 6. and leegend have been deleted.

Page 7 line 205: “At P5; DRD2 is relatively strongly expressed in Purkinje cell layer in the nax cerebellum”. This sentence does not match the one written in the legend of figure 3.

Answer: The sentence in the legend of figure 3 has been corrected according to reviewer valuable comment and (highlighted in yellow.

Page 7 line 210-211 “Similarly, RNAseq at P5/P7 revealed an increase in DRD2 gene expression in the nax cerebellum” is not completely true. Please correct with “ did not reveal a significant increase”.

Answer:  The statement in paragraph 3.2 has been fixed and highlighted in yellow.

It appears from figure 4 that there are no differences in DRD3 expression between nax and wt mice in deep cerebellar nuclei. (CN). Since 2 papers (Cutando et al., 2019; Locke et al., 2018) have recently identified DRD1-DRD2-expressing neurons in CN, it could be useful to check whether the CN, in addition to expressing DRD3, also express the DRD1 and DRD2 and thus verify whether differences exist between nax and wt mice.

Answer:  Thank you for your valuable suggestion, at p17, DRD1 was weakly expressed in both nax and wt cerebellar nuclei neurons which are similar and added on page 5 with data not shown, but it wasn’t the case in DRD2 and we couldn’t detect in cerebellar nuclei region.

In the nax cerebellum, DRD4 is expressed in a subset of PCs which is alternated with weak or lack of DRD4 expression similar to parasagittal stripes pattern (Figure 5). It would be very interesting to perform double immunostaining with zebrinII in order to reveal a possible array of alternating ZII/DRD4 parasagittal stripes in nax and wt cerebella.

Answer:  Thanks for reviewer valuable comment, this is a good points, and we are already considered it in more detail study. This experiment, and related ones, is on our to-do list, but the data will not be available for this MS.

Yours sincerely,

Hassan Marzban, Ph.D.

Associate Professor

Reviewer 2 Report

In this manuscript the authors describe the profile of  DRDs expression in the cerebellum of the Acp2 -/- (nax) mice. Previous publication from this lab was shown that these mutants had sever cerebellar defect particularly cerebellar cortex with alteration in some gene expression and impairment in compartmentalization. In this study DRDs expression in the nax and wt mice showed significant upregulations in Drd1, Drd3, and Drd4 gene expression in nax cerebella, but not Drd2 and Drd5 and authors concluded have no dependency on ACP2. The most interesting observation is DRD3 expression in a cells in granule cells that authors called them Golgi cells. This manuscript suffer from double staining and author could use a GC markers to characterize it, but I couldn’t see in figures. DRD4 stripe expression pattern in the nax cerebellum is interesting as well. It is not clear what these results mean with respect to cerebellar function, but the observations appear fairly solid and of interest to cerebellum researchers.  The manuscript is very well written, the image panels are masterfully constructed, and in total the mouse model/disease model they have chosen to study is very attractive. There are a major issues, which need to be addressed: The findings are extremely interesting, while at the same time may impress a marginal impact without adding an images to support last paragraph of the discussion.   Authors in final paragraph of discussion pointed out that mutated/misfolded Acp2 induce UPR and indirectly affect autophagy and lysosomal activity and downregulate lysosomal protein degradation and subsequently downregulate DRD degradation. Although this is an important point but there is no any data to support this notion. I think authors need to include an images regarding UPR or autophagy in nax model to support the final articulation of this study.

Minor comments

-        The authors state that "All animal procedures were based on institutional regulations...". This wording is too vague for such a sensitive subject as animal ethics. The authors have to state that all procedures were specifically approved for this project and name the ethic committee(s) who approved the application(s).

-        In all figures,  scale bars are presented unsuitably, please change them.

Author Response

Dr. Hassan Marzban

Department of Human Anatomy and Cell Science

College of Medicine, Faculty of Health Sciences

University of Manitoba

745 Bannatyne Avenue

Winnipeg, Manitoba, R3E 0J9

Canada

Editor-in-Chief

                                                                                                                    April 12, 2020

Dear Editor,

             We have revised the MS entitled:  “ALTERATION OF THE DOPAMINE RECEPTORS EXPRESSION IN THE CEREBELLUM OF THE LYSOSOMAL ACID PHOSPHATASE 2 MUTANT (NAKED-ATAXIA [NAX]) MOUSE” authored by Mehdi Mehdizadeh, Niloufar Ashtari, Xiaodan Jiao, Maryam Rahimi Balaei,  Asghar Marzban,  Farshid Qiyami Hour, Jiming Kong, Saeid Ghavami, and Hassan Marzban according to the reviewers' careful suggestions and comments. The details of the changes are as follows:

Reviewer #2: Comments and Suggestions for Authors

In this manuscript the authors describe the profile of DRDs expression in the cerebellum of the Acp2 -/- (nax) mice. Previous publication from this lab was shown that these mutants had sever cerebellar defect particularly cerebellar cortex with alteration in some gene expression and impairment in compartmentalization. In this study DRDs expression in the nax and wt mice showed significant upregulations in Drd1, Drd3, and Drd4 gene expression in nax cerebella, but not Drd2 and Drd5 and authors concluded have no dependency on ACP2. The most interesting observation is DRD3 expression in a cells in granule cells that authors called them Golgi cells. This manuscript suffer from double staining and author could use a GC markers to characterize it, but I couldn’t see in figures. DRD4 stripe expression pattern in the nax cerebellum is interesting as well. It is not clear what these results mean with respect to cerebellar function, but the observations appear fairly solid and of interest to cerebellum researchers.  The manuscript is very well written, the image panels are masterfully constructed, and in total the mouse model/disease model they have chosen to study is very attractive. There are major issues, which need to be addressed:

The findings are extremely interesting, while at the same time may impress a marginal impact without adding an images to support last paragraph of the discussion. Authors in final paragraph of discussion pointed out that mutated/misfolded Acp2 induce UPR and indirectly affect autophagy and lysosomal activity and downregulate lysosomal protein degradation and subsequently downregulate DRD degradation. Although this is an important point but there is no any data to support this notion. I think authors need to include an images regarding UPR or autophagy in nax model to support the final articulation of this study.

Answer:  This is a valid point and we have included our findings about the regulation of “Autophagy” by NAX mutation in the figure has been added as a supplement fig. and related information on page 12 (highlighted in yellow). Our results showed that ACP2 point mutation decreased LC3β lipidation, Atg5 and Atg12 conjugation, Beclin-1 expression, and Atg7 expression.

Minor comments

The authors state that "All animal procedures were based on institutional regulations...". This wording is too vague for such a sensitive subject as animal ethics. The authors have to state that all procedures were specifically approved for this project and name the ethic committee(s) who approved the application(s).

Answer:  It has been fixed

In all figures, scale bars are presented unsuitably, please change them.

Answer:  According to your valuable comment, the scale bars have been fixed.

Yours sincerely,

Hassan Marzban, Ph.D.

Associate Professor

Round 2

Reviewer 1 Report

Most of my concerns from my previous review have been addressed. I recommend an accept in the present form.